# 5′RNA *Trans*-Splicing Repair of *COL7A1* Mutant Transcripts in Epidermolysis Bullosa

**DOI:** 10.3390/ijms23031732

**Published:** 2022-02-02

**Authors:** Elisabeth Mayr, Michael Ablinger, Thomas Lettner, Eva M. Murauer, Christina Guttmann-Gruber, Josefina Piñón Hofbauer, Stefan Hainzl, Manfred Kaiser, Alfred Klausegger, Johann W. Bauer, Ulrich Koller, Verena Wally

**Affiliations:** 1EB House Austria, Research Program for Molecular Therapy of Genodermatoses, Department of Dermatology and Allergology, University Hospital of the Paracelsus Medical University, 5020 Salzburg, Austria; el.mayr@salk.at (E.M.); m.ablinger@salk.at (M.A.); thomas.lettner@gmail.com (T.L.); e.murauer@salk.at (E.M.M.); c.gruber@salk.at (C.G.-G.); j.d.pinon@salk.at (J.P.H.); s.hainzl@salk.at (S.H.); manfred_kaiser@outlook.com (M.K.); a.klausegger@salk.at (A.K.); joh.bauer@salk.at (J.W.B.); 2Department of Dermatology and Allergology, University Hospital of the Paracelsus Medical University, 5020 Salzburg, Austria

**Keywords:** *COL7A1*, epidermolysis bullosa, RNA therapy, RNA *trans*-splicing

## Abstract

Mutations within the *COL7A1* gene underlie the inherited recessive subtype of the blistering skin disease dystrophic epidermolysis bullosa (RDEB). Although gene replacement approaches for genodermatoses are clinically advanced, their implementation for RDEB is challenging and requires endogenous regulation of transgene expression. Thus, we are using spliceosome-mediated RNA *trans*-splicing (SMaRT) to repair mutations in *COL7A1* at the mRNA level. Here, we demonstrate the capability of a *COL7A1*-specific RNA *trans*-splicing molecule (RTM), initially selected using a fluorescence-based screening procedure, to accurately replace *COL7A1* exons 1 to 64 in an endogenous setting. Retroviral RTM transduction into patient-derived, immortalized keratinocytes resulted in an increase in wild-type transcript and protein levels, respectively. Furthermore, we revealed accurate deposition of recovered type VII collagen protein within the basement membrane zone of expanded skin equivalents using immunofluorescence staining. In summary, we showed for the first time the potential of endogenous 5′ *trans*-splicing to correct pathogenic mutations within the *COL7A1* gene. Therefore, we consider 5′ RNA *trans*-splicing a suitable tool to beneficially modulate the RDEB-phenotype, thus targeting an urgent need of this patient population.

## 1. Introduction

Dystrophic epidermolysis bullosa (DEB) is caused by mutations in the *COL7A1* gene. Its product, type VII collagen (C7), a 290 kDa protein, is the major constituent of basement membrane anchoring fibrils and is therefore crucial for the maintenance of skin integrity [1]. *COL7A1* comprises 118 exons encoding an mRNA transcript of 9.2 kb [2]. *COL7A1* mutations can have a wide spectrum of clinical consequences, ranging from skin fragility to the involvement of mucous membranes. Additionally, various secondary and extracutaneous manifestations [3], such as excessive itch, wound chronification and the development of particularly aggressive squamous cell carcinomas are common [1]. In contrast to dominant mutations, which mainly result in glycine substitutions, recessively inherited mutations, such as nonsense mutation, splice site alterations or indels, lead to a significant reduction or loss of C7 expression. In contrast to other types of EB (e.g., EBS), where mutational hotspots are common, in RDEB, mutations along the entire coding sequence of *COL7A1* have been described [1,4].

Phenotypic correction of RDEB can be achieved by stable substitution of the missing protein using gene-, cell-, protein- or systemic therapy approaches. Most therapeutic efforts in EB have entailed gene replacement therapy by targeting patient keratinocytes or fibroblasts in an ex vivo gene therapy approach [5,6,7,8,9,10]. In these cases, epidermal stem cells from patients were transduced in vitro, expanded into skin sheets and then retransplanted onto the patients. Woodley et al. developed a type VII collagen mini-gene containing the intact non-collagenous domains NC1 and NC2 and part of the central collagenous domain that directed persistent synthesis and secretion of a 230 kDa recombinant minicollagen VII after transduction into DEB keratinocytes [11]. Reversion of the DEB phenotype was demonstrated by several approaches focusing on integration of full-length *COL7A1* by using different strategies, including lentiviral transduction [12], cosmid clones [13], microinjection of a P1-derived artificial chromosome (PAC) [14] and retroviral transduction [8,9]. Titeux et al. described an approach in which *COL7A1* cDNA under the control of a human promoter and incorporated into a minimal self-inactivating retroviral vector, enabled ex vivo genetic correction of *COL7A1* in RDEB keratinocytes and fibroblasts by full-length *COL7A1* insertion, and generated long-term expression of recombinant type VII collagen; as a result, the dermal–epidermal adherence in generated skin equivalents was restored [6]. Similarly, Siprashvili et al. successfully obtained long-term *COL7A1* expression in regenerated human RDEB epidermal xenografts upon transduction of patient keratinocytes with a Moloney leukemia virus-derived retroviral vector carrying the respective cDNA [7]. Treatment of six wounds in a total of four RDEB patients with epidermal sheets expanded from genetically corrected keratinocytes showed improved wound healing and restored type VII collagen expression [8]. Further, anchoring fibrils were detected in biopsies of treated skin areas at the dermal-epidermal basement membrane. However, the decline of type VII collagen expression within the basement membrane zone (BMZ) over time revealed a correlation between type VII collagen expression within the BMZ and the observed decrease in clinical improvement [9,15]. This can be explained by transduction difficulties, the random integration of the transgene flanked by viral sequences, the large size of *COL7A1* cDNA, aberrant splicing, or the number of engrafted epidermal stem cells [6,15,16,17].

Thus, we previously presented an alternative way to correct defects in the 3’ coding region of *COL7A1* by RNA *trans*-splicing, potentially reducing the risk of genetic rearrangements of the large repetitive cDNA sequence [18,19,20]. *Trans*-splicing is a naturally occurring process in which exons from two or more different pre-mRNAs are combined to produce one mature RNA molecule [21]. For therapeutic purposes, an engineered RNA *trans*-splicing molecule (RTM) induces the *trans*-splicing process to replace a 3′, 5′ or internal sequence of an endogenously expressed target pre-mRNA, thereby generating a new chimeric mature RNA [22]. *Trans*-splicing offers several advantages over conventional gene replacement therapies [23]. First, the size of the transgene can be reduced. Second, expression of the transgene is regulated by control elements of the endogenous target gene because the transgene is specifically spliced into the endogenous transcript. Third, off-target effects due to unintended delivery or misregulation are minimized, as *trans*-splicing should occur only in those cells naturally expressing the target pre-mRNA. Finally, RNA *trans*-splicing offers the potential to convert dominant negative mutants into wild-type gene products by decreasing the amount of mutated transcripts and thus the dominant negative effect [24,25,26]. Overall, the efficiency and functionality of the SMaRT technology has been shown in several model systems in vitro and in vivo [18,27,28,29,30,31,32,33], including mutations in genes coding for factor VIII (FVIII) [32], cystic fibrosis transmembrane conductance regulator [31], β-globin [27], DNA protein kinase catalytic subunit [29], the dystrophia myotica protein kinase and survival motor neuron protein (SMN2) [34]. Proof of RNA *trans*-splicing functionality in a skin-specific background was first obtained by Dallinger et al. for the *COL17A1* gene [35]. Additionally, 5′ *trans*-splicing-mediated transient correction of mutations in the *PLEC* [28] and *K**RT**14* [24,25,26] genes was reported. Finally, we recently showed the feasibility of SMaRT to correct *COL7A1* mutations in RDEB patient cells using 3′*trans*-splicing [18,19,20]. Both retroviral [18] or lentiviral [20] transduction of RDEB keratinocytes with a 3′RTM resulted in an increase of *COL7A1* transcript levels leading to the restoration of C7 expression. In addition, a normal localization of C7 within the BMZ of skin equivalents was shown and the restored protein assembled into anchoring fibril-like structures, thus demonstrating the potential of RNA *trans*-splicing to correct the RDEB phenotype in vitro [18]. However, these approaches were restricted to *COL7A1* exons 47–118 [19] and exons 65–118 [20], thus covering mutations in the 3′ portion of the transcript. Therefore, here we present an approach to correct mutations in the 5′ part of *COL7A1* (including exons 1–64), thus enabling correction of all dominant and recessive *COL7A1* mutations with only two designed RTMs (5′ and 3′).

## 2. Results

### 2.1. Selection of Highly Functional RTMs Using a Fluorescence-Based Screening System

To identify binding domains (BDs) that induce the *trans*-splicing reaction with high efficiency, we performed a fluorescence-based RTM screen that we have previously established and successfully applied for the development of several correcting RTMs targeting various EB-associated genes [22,25,36,37,38,39]. To accomplish this, a reporter mini-gene target (Tr) was engineered containing *COL7A1* exon/intron 64 and a 3′ portion of AcGFP, mimicking *COL7A1* exons 65–118. The reporter RTMs (RTMrs) contained: (a) a full-length DsRed as a transfection reporter, linked with (b) the 5′ portion of AcGFP (mimicking *COL7A1* exons 1–64), (c) a functional 5′ splice site (d) a short spacer region and (e) one or more randomly cloned BDs reverse complementary to the target region (Figure 1A). Individual RTMrs with BDs complementary to *COL7A1* exon/intron 64 were characterized by sequence analysis and co-transfected with the mini-gene target into HEK293FT cells. Additionally, plasmids, expressing the DsRed-AcGFP fusion protein, were transfected into HEK293FT cells, revealing a general transfection efficiency of >90% (Figure 1B). Treated cells were analyzed for their reporter gene expression derived from accurate *trans*-splicing by fluorescence microscopy and flow cytometric analysis (Figure 1B, Appendix A). As the number of DsRed- and AcGFP-expressing cells and the intensity of the AcGFP signal correlate with the *trans*-splicing efficiency of the introduced RTM, we evaluated the *trans*-splicing efficiency of single RTMrs by calculating the AcGFP/DsRed ratios in co-transfected HEK293FT cells. RTMrA and RTMrB were identified as the most efficient repair molecules, both inducing a AcGFP/DsRed ratio of >90% (Figure 1B,C). Respective binding domains were 462 bp (A) and 312 bp (B) long and reverse-complementary to, respectively, three and two *COL7A1* intron 64 internal pre-mRNA sequence stretches.

### 2.2. Endogenous 5′ Trans-Splicing in Keratinocytes

To evaluate RNA *trans*-splicing in a skin context, we transiently transfected RTMrA and RTMrB into HaCaT keratinocytes. Accurate *trans*-splicing led to the fusion of the 5′AcGFP, derived from the RTMr, with the endogenous *COL7A1* pre-mRNA (exons 65–118). The resulting GFP-*COL7A1* fusion product was verified by PCR analysis using a specific forward primer for AcGFP and a specific reverse primer for *COL7A1* and a subsequent nested PCR. Sequence analysis confirmed correct *trans*-splicing (Figure 1D,E).

### 2.3. Endogenous Correction of COL7A1 in RDEB Patient Cells

To quantify levels of recovered *COL7A1* mRNA, the 5′AcGFP sequence of the RTM backbone was substituted by the wild-type coding sequence of *COL7A1*, spanning from exon 1 to exon 64. After retroviral delivery of the endogenous RTMs (RTMe) into RDEB patient keratinocytes compound heterozygous for the mutations p.R578X (exon 13) and c.7786delG (exon 104), the mutation in exon 13 was replaced by the wild-type sequence provided by the RTM, resulting in one full-length and functional wild-type allele. Overall, expression of *COL7A1* was analyzed by sqRT-PCR and showed that transcript levels in RTMeA- and RTMeB-transduced RDEB patient keratinocytes (RDEB-KC) increased significantly (*p* < 0.05) over controls (i.e., untransfected RDEB-KCs, mock-transfected RDEB-KCs) for both RTMeA (2.6-fold, corresponding to 75% wild-type levels) and for RTMeB (1.8-fold, i.e., 51% of wild-type levels) (Figure 2A).

At the protein level, Western blot analysis of supernatants of RDEB patient keratinocytes stably expressing RTMeA or RTMeB proved reconstitution and correct secretion of C7 (Figure 2B). In detail, while wild-type keratinocytes showed a clear C7-specific band at the expected size of 290 kD, RDEB-KC and mock control secreted only negligible C7 levels. In contrast, patient keratinocytes transduced with RTMes showed a clear re-expression and secretion of C7. Additional bands appearing at about 200 kDa, detected in RTM-transduced cells only, may indicate direct expression of the unspliced RTMe. 

To further prove re-expression of C7 at the protein level, immunofluorescence staining was performed on RTM-transduced patient keratinocytes. Human wild-type keratinocytes served as a positive control and showed the characteristic cytoplasmic C7 staining. The RDEB-KC cell line showed less intensive staining, with the remaining potentially mutant C7 accumulated around the nucleus. In contrast, RTM-transduced patient keratinocytes showed a C7 expression pattern comparable to that of wild-type keratinocytes (Figure 3A).

### 2.4. C7 Expression in Skin Equivalents

To determine whether recovered C7 localized correctly in a tissue context, we generated skin equivalents (SE) from RDEB-KC transduced with RTMeA, as the more efficient of the two RTMs. SEs derived from RTMeA corrected keratinocytes showed a C7 signal at the level of the BMZ, which was absent in SEs derived from patient keratinocytes transduced with the empty vector (mock). SEs generated from wild-type keratinocytes also showed a clear and characteristic C7-staining at the level of the BMZ. These results revealed the correct processing and localization of the *trans*-spliced C7 (colocalized with Laminin 332) within the BMZ of skin equivalents (Figure 3B, Appendix A).

## 3. Discussion

Curative therapy for any form of EB is still not available. Here we demonstrate the feasibility of the SMaRT approach to correct the complete 5′ portion of *COL7A1*, corresponding to 5487 nt of the transcript, with a single molecule. Given the heterogeneity of mutations observed in RDEB, this, together with a previously published RTM correcting the remaining 3′ *COL7A1* portion [18], now covers any mutation occurring in RDEB. Successful correction was shown at both the RNA and protein level, as well as functionally in 3-D skin equivalents. This advance, combined with our recently described approach to target mutant sites in exons 65 to 118 [18], now enables us to correct the entire *COL7A1* transcript with only two therapeutic molecules.

SMaRT is a promising approach in the field of RNA therapies, offering many advantages particularly for large transcripts like *COL7A1*. Although other approaches have proven successful to restore *COL7A1* expression as well, they come with certain bottlenecks that do not apply for the *trans*-splicing approach. For example, antisense oligonucleotides (ASOs) were recently shown to mediate the skipping of a specific mutated exon [40,41], resulting in the expression of slightly truncated but functional C7. However, this approach can only be applied for in-frame exons that are not located in functionally important regions of the protein. On the other hand, retroviral transduction of full-length *COL7A1* lacks controlled expression at endogenous levels, thus potentially bearing the risk of side effects due to overexpression. In addition, the high sequence repetitiveness and large size of the *COL7A1* cDNA (~9.2 kb) increases the risk of genetic rearrangements of the transgene after its virus-mediated integration into the host’s genome [6]. First attempts to stably introduce the *COL7A1* cDNA into the genome of patient’s keratinocytes had limited success. In a recently conducted phase I/II clinical trial, including seven patients with RDEB, improved wound healing and anchoring fibril formation at the graft sites was achieved. Unfortunately, type VII collagen expression levels significantly decreased over time [8,9]. The study showed a persistence of type VII collagen expression in two patients 2 years after treatment. Reasons for that could be a low virus transduction efficiency, the large size of the *COL7A1* transgene, aberrant splicing, or a low targeting rate of long living holoclones [6,9,15,16,17]. Currently there are several ongoing clinical trials based on the correction of keratinocytes via gene replacement for EB (www.clinicaltrials.gov: NCT03490331, NCT04227106, NCT02984085, NCT01874769). Taken together, while several strategies have been developed to replace or correct C7 in dystrophic EB patients, they usually come with certain restrictions and disadvantages that need to be considered, e.g., for individual patients and different genotypes. RNA *trans*-splicing will integrate well in a panel of therapy options, coming with a potentially high safety profile. However, prior to a clinical application it will be necessary to confirm safety and specificity of the *trans*-splicing approach in in vivo models and in clinics. 

We showed that RTM-transduced RDEB-KC expressed *COL7A1* mRNA at a level slightly below that of wild-type keratinocytes, which can be explained by the fact that only one mutation was located upstream of exon 64 (Figure 2). More importantly, we showed restoration of C7 on the protein level and the correct localization of the protein within the BMZ in skin equivalents.

In conclusion, we present here the first proof-of-principle data for stable, functional 5′*trans*-splicing for RDEB patients, overcoming limitations of cDNA gene therapy for the large *COL7A1* gene. Even though the here presented RNA *trans*-splicing approach does not provide correction at the genomic level, the fact that C7 has a half-life of at least one month [42] renders attractive a transient correction at sites presenting with particularly burdensome lesions. Although repeated treatments are needed, this will potentially go along with a favorable safety profile. In this context, delivery will be a key aspect to be addressed in the future. 

Having now available both 3′ and 5′ repair molecules targeting the same intron gives us the potential to correct all possible mutations in the *COL7A1* gene, regardless of their mode of inheritance. This approach could be suitable for ex vivo or in vivo RNA therapy of all patients with DEB. 

## 4. Materials and Methods

### 4.1. Cloning of Screening Constructs

All constructs for the 5′ RTM screen were cloned into the pcDNA3.1D/V5-HIS-TOPO vector backbone (Invitrogen, Carlsbad, CA, USA). The reporter mini-gene target (Tr) contained the exon/intron 64 region of *COL7A1* and the 3′ portion of AcGFP. Exon 64 and intron 64 of human *COL7A1* were amplified from genomic wild-type DNA of healthy keratinocytes using Pfu-Turbo polymerase (Stratagene, La Jolla, CA, USA). Primers used (fw: 5′ctagaagcttggaaagccaggcgagg 3′, rv: 5′ctagggatccagttagtagggcagaggactcacatcag 3′) included HindIII (fw) and BamHI (rv) restriction sites for cloning. For 3′ AcGFP amplification, a specific primer pair (fw: 5′tatggatccggcgcaggagccggcgccaccatggtgagcaagggcgc 3′ rv: 5′ gatctctagatcacttgtacagctcatc 3′) was used for amplification from the pAcGFP vector (Clontech, Mountain View, CA, USA). Positive clones were verified by Sanger sequencing. All constructs were sequenced with an ABI Prism automated sequencer 3130 using an ABI PRISM dye terminator cycle sequencing kit (Applied Biosystems, Foster City, CA, USA). 

The RTM backbone contained a full-length DsRed as a transfection control, a linker sequence (5′ggagcaggcgccggatccggcgcaggagccggcgccacc 3′) and the 5′ portion of AcGFP. For amplification of the reporter genes, a DsRed-specific primer pair (fw: 5′gatcaagcttcaccatggacaacaccgaggacgt 3′, rv: 5′tatgatatcggatccggcgcctgctccctgggagccggagtggcgg 3′) and a 5′ AcGFP-specific primer pair (fw: 5′tatggatccggcgcaggagccggcgccaccatggtgagcaagggcgc 3′, rv: 5′atagatatctgtaataatatcgcaacgagctctcttacctcggcgcgcga 3′) were designed. pAcGFP and the pDsRed plasmids (Clontech, Mountain View, CA, USA) were used as templates. Downstream of the spacer region, randomly generated BDs, created by sonication or restriction digest of a purified (GFX gel purification, GE Healthcare, Buckinghamshire, UK) *COL7A1* exon / intron 64 PCR product, were cloned into the EcoRV restriction site. 

Upon transformation into TOP10 chemically competent cells (Invitrogen, Carlsbad, CA, USA), bacterial clones carrying an RTM with a distinct BD were identified by colony PCR using Firepol polymerase I (Solis Biodyne, Tartu, Estonia) and vector-specific primers (fw: 5′gctgaccctgaagttcatctg 3′, rv: 5′caccccaccccccagaatag 3′). Approximately 50 individual clones were analyzed for RTM intake by sequence analysis. RTMs containing BDs with sequences complementary to the target region (exon/intron 64) were considered for further experiments. The sequences of the two most efficiently performing, and thus further evaluated BDs, were: 

RTM A: 5′gatttgtgtgtgtcttggagcatggcctgtggccgtctgagtgagctgctacatgtctaagggttagcaacacatgaggcacacgtgtagacacatgcatcctggcccagacatacagatgcatccagaaacatagttttttttttttaggaggccgaggcgggcggatcacctgaggtcagtttgtgatcagcctgactaacatggagaaaccccagtgcatatccatgtgtgtcttagtacatgtacccatgggtctgtgtgtcctggtctgtttgcacatcttggcccatgtctgcatatgggcatgggcttatacatgtctccaaagagcatgggcttatacatgccttgagagccatgtgtcttggcatctgtctgcaactcttgtgtgtagctcaatgtgtatgaccataagcgtgtgggtgaccatataaagtcatgggccagcatgtcctagca 3′ 

RTM B: 5′tgtgactgtatttctgtgcctgtagctccatggttgtgtgtgtcttggagcatggcctgtggccgtctgagtgagctgctacatgtctaagggtgtgcctgcataggcctgctcaccccaacaggcaactgcacacatgtcctttgactctggctctctagaggtagctcccgcaatggctcacgcctgtaatcccagcactttaggaggccgaggcgggcggatcacctgaggtcagtttgtgatcagcctgactaacatggagaaaccccatctctactaaaaatacaaaattagccaggcatggtggcg 3′

Controls: Positive control reporter-based: full-length DsRed and full-length AcGFP, connected by the linker sequence and cloned under the control of a CMV promoter. 

Negative controls: RDEB-KC mock transduced with pMX vector.

### 4.2. Plasmid Transfection and Trans-Splicing Evaluation

For transient (co-)transfection of HEK293FT (human embryonic kidney cells, ATTC) cells with the screening constructs, jet-PEI reagent (Polyplus-transfection SA-BIOPARC, Illkirch, France) was used. Transfection was performed according to the manufacturer’s protocol in 6-well plates. 

Cells double positive for AcGFP and DsRed were considered to show successful *trans*-splicing. Visualization was done using the epifluorescence Zeiss Axiophot microscope (Carl Zeiss, Oberkochen, Germany).

In addition, HEK293FT cells co-transfected with a target molecule and an individual RTM were analyzed for their reporter gene expression by flow cytometry. Cells seeded in 6-well plates were washed with Dulbecco’s PBS (Biochrom, Berlin, Germany), trypsinized and centrifuged at 180× *g* for 5 min. The cell pellet was resuspended in 1 mL PBS and ~50,000 cells were analyzed using a Beckman Coulter FC500. Data were analyzed using the CXP software. 

For the analysis of endogenous *trans*-splicing in keratinocytes, transfections were performed using the electroporation-based Amaxa Nucleofector (Lonza, Basel, Switzerland) technology. Prior to transfection, 4 × 10^6^ keratinocytes were trypsinized, pelleted and resuspended in 100 μL transfection solution V (according to the manufacturer’s protocol), containing 1 μg plasmid DNA and pulsed with Amaxa program U-20 (according to the manufacturer’s protocol). After transfection, cells were carefully dispensed into 37 °C medium and seeded in 6-well plates. 

### 4.3. RNA Isolation

To detect GFP-*COL7A1* mRNA fusion products, RNA was isolated 4 days post transfection using the RNeasy Mini Kit (Qiagen, Hilden, Germany) according to the manufacturer’s protocol.

### 4.4. cDNA Amplification and PCR

cDNA from 1 μg total RNA was synthesized using the iScript cDNA Synthesis Kit (Bio-Rad, Munich Germany) according to the manufacturer’s protocol.

To detect GFP-*COL7A1* fusion transcripts as proof of accurate endogenous *trans*-splicing between the RTM and the endogenous target pre-mRNA, nested PCR using a AcGFP-specific forward (fw: 5′ atcctgatcgagctgaatgg 3′, fw: 5′ gctgaccctgaagttcatctg 3′), and *COL7A1*-specific reverse (5′ctgggacaccaggaaaacc 3′, 5′ gggtccaaggataccaggag 3′) primers was performed. 

PCR bands were analyzed via gel electrophoresis, purified by GFX gel purification (GE Healthcare, Buckinghamshire, UK) and subsequently analyzed by Sanger sequencing as described above. 

### 4.5. Cloning of Constructs for Endogenous Experiments

Due to its length, the coding sequence of *COL7A1* spanning from exon 1 to 64 was split into three fragments (F) and each portion was PCR-amplified using Pfu Turbo Polymerase (Agilent Technologies, Santa Clara, CA, USA) and a cDNA of a wild-type keratinocyte as template. Fragments were selected according to the presence of unique, naturally occurring restriction sites at the ends of the fragments. Three different primer pairs (F1: fw: 5′ctaggaattccaccatgacgctgcggcttct 3′; rv: 5′ctaggcccgggcagctcggtggcttgcag 3′, F2: fw: 5′ctaggcccgggcagcgggtgcgagtgtcc 3′; rv: 5′ ctagttaagcccagaagcctgggc 3′, F3: fw: 5′ gggcacagccgtggtc 3′; rv: 5′ctaggcggccgctgtaataatatcgcaacgagctctcttacgttttttccattcaggccag 3′) were designed to amplify all three *COL7A1* fragments. A functional 5′ splice site (ag/gtaaga) and a short spacer sequence (5′gtaagagagctcgttgcgatattattaca 3′) were introduced by the reverse primer for F3 amplification. All PCR products were initially cloned into a Strataclone vector (Agilent Technologies, Santa Clara, CA, USA) according to the manufacturer’s protocol. Meanwhile, additional restriction sites (SrfI and AleI) were inserted into the multiple cloning site (MCS) of the target vector pMx-IRES-Blasticidin by inserting a designed oligonucleotide (5′gaattcctaggcccgggcctagcacagccgtgctagctcgag 3′) into the vector using the restriction sites for EcoRI and XhoI. *COL7A1* F1 was cloned in between EcoRI and SrfI, F2 in between SrfI and AleI, and F3 in between AleI and NotI.

The most efficient BD from RTMAe identified in the screen was amplified using GoTaq Polymerase (Promega, Mannheim, Germany) according to the manufacturer’s protocol and using primers that included NotI restriction sites (BD A: fw: 5′ ctaggcggccgcgatttgtgtgtgtcttggagcatg; rv: 5′ ctaggcggccgctgctaggacatgctggccc 3′ BD B: fw: 5′ctaggcggccgcgatgtgactgtatttctgtgcctgtag 3′; rv: 5′ ctaggcggccgccgccaccatgcctggc 3′). The BD was digested with NotI and cloned into NotI-digested pMx-IRES-Blasticidin Vector containing exons 1–64 of human *COL7A1*, a functional 5′ splice site and a short spacer region. Correct orientation of the BD was determined by sequence analysis.

### 4.6. Viral Transduction

Phoenix packaging cells were seeded in Dulbecco’s Modified Eagle’s Medium (DMEM) containing 10% FCS, transfected at a confluency of 60% with 8 μg retroviral plasmid DNA using jet-PEI transfection reagent (Polyplus-transfection SA, Illkirch-Graffenstaden, France) and incubated overnight at 37 °C. Twenty-four hours post transfection the medium was changed and the cells were incubated at 32 °C. Viral supernatants were collected every 8 h between 48 h up to 96 h and stored at 4 °C. After filtration through a 22 μm cell strainer, 5 μg/mL polybrene (Sigma, St. Louis, MO, USA) was added. Target cells were infected at a confluency of 50% by adding the viral supernatant directly into the flask. Viral particles were spun onto cell surfaces by centrifugation at 600× *g* for 90 min at 32 °C. Cells were incubated overnight at 32 °C, washed four times with Dulbecco’s PBS (Biochrom, Berlin, Germany) and incubated for two days in in the presence of primozin (Invivogen, San Diego, CA, USA) (25 mg/L). Selection for successful transduction was done using blasticidin (Thermo Fisher Scientific, Waltham, MA, USA) for 8 days at a concentration of 200 mg/L.

### 4.7. Semi-Quantitative (sq)RT-PCR

For sqRT-PCR, a Bio-Rad CFX96 Real-time system was used to relatively quantify transcript levels of *COL7A1*. *GAPDH* was used as reference. Primers: *COL7A1* (Ex-61/62-fw: 5′ tgggccgaatggtgctgca 3′; Ex-65/66-rv: 5′ ctttctctcccttcctcccg 3′); *GAPDH* (*GAPDH*-fw: 5′gccaacgtgtcagtggtgga 3′; *GAPDH*-rv: 5′caccaccctgttgctgtagcc 3′). 

qPCR was performed using the SybrGreen Super Mix (Bio-Rad, Munich, Germany) according to the manufacturer’s protocol, using 12.5 ng cDNA as template. Cycling conditions were 3 min at 95 °C followed by 40 cycles of 20 s at 95 °C, 20 s at 65 °C and 30 s at 72 °C followed by a melt curve analysis spanning 65 °C to 95 °C. *COL7A1* expression was calculated using ∆CT method. In addition, PCR products were analyzed on a 1% agarose gel and by sequence analysis to prove specificity. All reactions were performed in duplicate and standard deviations were calculated from at least three independently performed experiments.

### 4.8. Western Blot Analysis

5 × 10^4^ cells were seeded and cultured for 4 days until 60% confluence in the presence of 50 μg/mL ascorbic acid to allow full prolyl- and lysyl-hydroxylation of newly synthesized collagens. Visual control of confluence is critical as clustering enhances type VII collagen expression significantly. Supernatants were collected and concentrated using Amicon ultra centrifugal filters with a cut-off of 10 kD (Millipore, Billerica, MA, USA). Complete Mini protease inhibitor cocktail tablets (Roche, Basel, Switzerland) were used to inhibit proteases according to the manufacturer’s protocol. Proteins were separated on a denaturing NuPAGE 4–12% Bis-Tris gradient gel (Thermo Fisher Scientific, Waltham, MA, USA) according to the manufacturer’s protocol at 150 V for 120 min, and transferred to a nitrocellulose membrane (Hybond-TM-ECLTM, Amersham Biosciences, Buckinghamshire, UK) by electroblotting for 75 min at 0.25 A. The membrane was blocked for 1 h at room temperature with 1% blocking buffer (Roche, Basel, Switzerland). Human type VII collagen was detected with anti-collagen type VII rabbit polyclonal antibody (Calbiochem, San Diego, CA, USA) in 0.5% blocking buffer (Roche, Basel, Switzerland) as first antibody overnight at 4 °C. HRP-labeled EnVision + anti-rabbit antibody (Dako, Vienna, Austria) in 0.5% blocking buffer (Roche, Basel, Switzerland)was used as the secondary antibody and incubated for 1 h at room temperature. Bands were visualized with an Immun-Star Western C Chemiluminescent Kit (Bio-Rad, Munich, Germany) and ChemiDoc XRS Imager (Bio-Rad, Munich, Germany).

### 4.9. Immunofluorescence Staining of Type VII Collagen in Keratinocytes

3 × 10^4^ cells were seeded into chamber slides and fixed for 20 min at –20 °C with methanol, washed once with PBS and blocked at room temperature for one hour with 2% BSA in PBS. Type VII collagen was detected with anti-collagen type VII rabbit polyclonal antibody (Calbiochem, San Diego, CA, USA) (1:200) in 2% BSA in PBS overnight at 4 °C. Second-step controls were not treated with the first antibody.

As a secondary antibody we used Alexa-Fluor-488 goat anti-rabbit (Thermo Fisher Scientific, Waltham, MA, USA) (1:400) in PBS for one hour, followed by DAPI staining (VWR, Vienna, Austria). After mounting, cells were analyzed using an epifluorescence Zeiss Axiophot microscope (Carl Zeiss, Oberkochen, Germany). Exposure time was optimized for the wild-type control and kept equal for all samples.

### 4.10. Generation of Skin Equivalents

3D skin equivalents (SE) were generated as described by Murauer et al. [18]. Collagen type VII-deficient fibroblasts were embedded in fibrin gel matrix and immersed in DMEM for 24 h at 37 °C in a humidified atmosphere and 5% CO_2_. Keratinocytes (6 × 10^5^ cells) were seeded onto the matrix and grown to confluence in Green medium [38]. At confluency, SEs were then lifted to the air-liquid interface to allow differentiation for 28 days. SEs were embedded in optimal cutting temperature (O.C.T) compound (Sakura Finetek, Staufen im Breisgau, Germany), frozen in liquid nitrogen, and cut into 6 μm sections for immunofluorescence staining.

### 4.11. Immunofluorescence Staining of Type VII Collagen and Laminin 332 in Skin Equivalents

Sections were fixed with acetone at 4 °C for 20 min and subsequently treated the same way as described for cells above. Laminin 332 antibody from R&D Systems (MAB21441) was used at 2 μg/mL.

### 4.12. Cells

HEK293FT cells (human embryonic kidney cells, ATTC) were grown at 37 °C and 5% CO_2_ in a humidified incubator in DMEM (Biochrom, Berlin, Germany) supplemented with 10% FCS (Biochrom, Berlin, Germany) and 100 U/mL penicillin/streptomycin (Biochrom, Berlin, Germany) and passaged every 4–5 days.

HaCaT cells (human spontaneously immortalized keratinocytes, ATTC) were grown at 37 °C and 5% CO_2_ in a humidified incubator in Epilife (Cascade Biologics, Portland, OR, USA) supplemented with HKGS (human keratinocyte growth supplement) and CaCl_2_ according to the manufacturer’s protocol and 100 U/mL penicillin/streptomycin (Biochrom, Berlin, Germany) and passaged every 4–5 days.

Primary wildtype keratinocytes were E6/E7-immortalized and grown at 37 °C and 5% CO_2_ in a humidified incubator in Epilife (Cascade Biologics, Portland, OR, USA) supplemented with HKGS (human keratinocyte growth supplement) and CaCl_2_ according to the manufacturer’s protocol and 100 U/mL penicillin/streptomycin (Biochrom, Berlin, Germany) and passaged every 4–5 days.

Primary keratinocytes were derived from an RDEB patient heterozygous for the mutations p.R578X and c.7786delG in exons 13 and 104 of the *COL7A1* gene (kindly provided by Prof. Meneguzzi, INSERM, Nice, France, [18]) and E6/E7 immortalized. Cells were grown at 37 °C and 5% CO_2_ in a humidified incubator in Epilife (Cascade Biologics, Portland, OR, USA) supplemented with HKGS (human keratinocyte growth supplement) and CaCl_2_ according to the manufacturer’s protocol and 100 U/mL penicillin/streptomycin (Biochrom, Berlin, Germany) and passaged every 4–5 days.

Primary fibroblasts were derived from an RDEB patient homozygous for the mutation c.258delAG of the *COL7A1* gene and E6/E7 immortalized. Cells were grown at 37 °C and 5% CO_2_ in a humidified incubator in DMEM supplemented with 10% FCS (Biochrom, Berlin, Germany)and 100 U/mL penicillin/streptomycin (Biochrom, Berlin, Germany) and passaged every 4–5 days.

## 5. Patents

JW Bauer is inventor on a patent on *trans*-splicing-Improved pre mRNA *trans*-splicing molecule (RTM) molecules and their uses. EU: 1402/2320952; US: 8,735,366; Japan: 5735912.

## Figures and Tables

**Figure 1 ijms-23-01732-f001:**
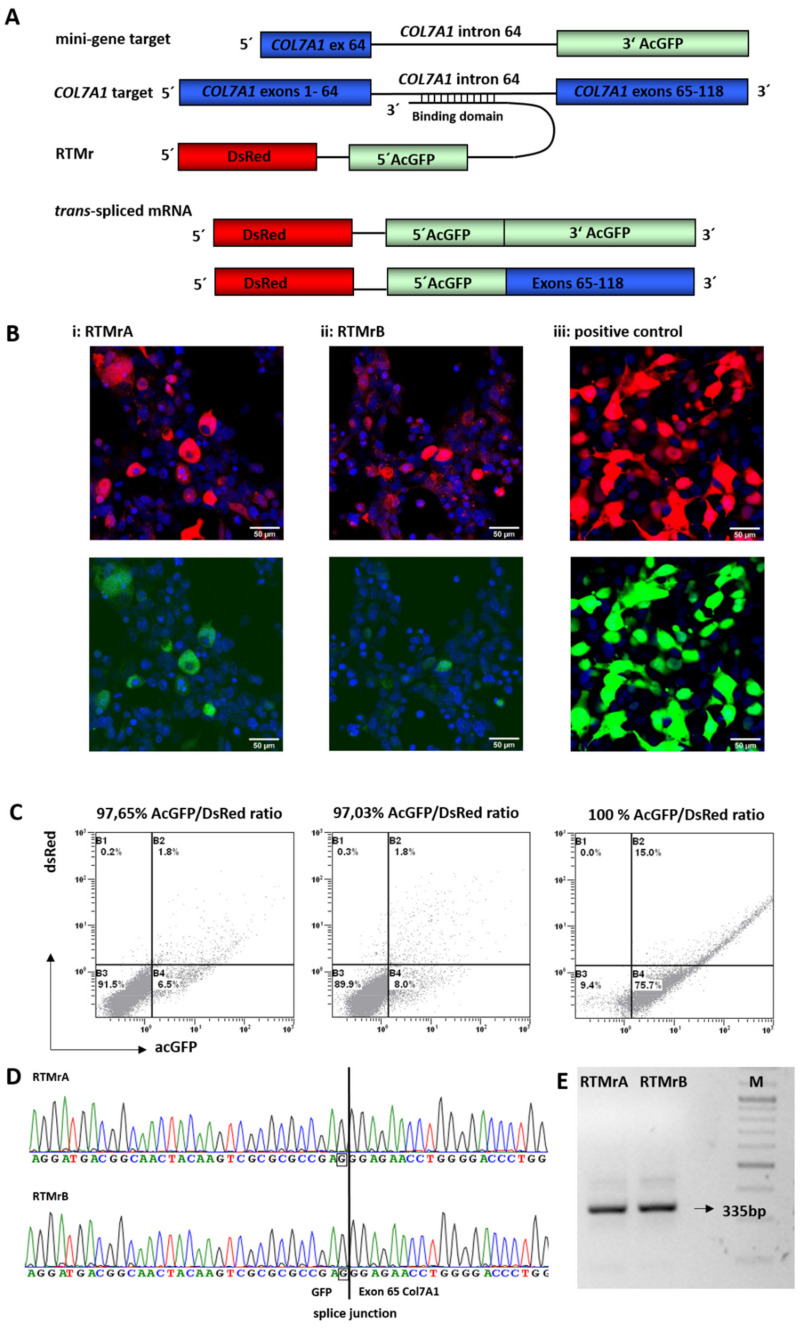
Fluorescence-reporter based screening for most functional RTMs. (**A**) *Trans*-splicing was evaluated either in the presence of a mini-gene target for screening of highly functional BDs, or in keratinocytes, where successful *trans*-splicing to the endogenous *COL7A1* target pre-mRNA was investigated. In the presence of a functional BD, specific *trans*-splicing results in a single mRNA transcript encoding either DsRed (transfection control) and full-length AcGFP in the mini-gene reporter setting, or a 5′AcGFP/*COL7A1* exons 65–118 hybrid when splicing to the endogenous transcript. (**B**,**C**) HEK293FT cells co-transfected with the mini-gene target and individual RTMrs (i,ii) and positive control (iii). Hybrid DsRed and AcGFP expression is shown by fluorescence microscopy and flow cytometry. AcGFP/DsRed ratios were calculated from the amount of GFP-positive cells (sector B2 + B4)/total amount of transfected cells (sector B1 + B2 + B4), and is given in percent (%). RTMrA showed an AcGFP/DsRed ratio of 97.65%; RTM B, 97.03%. 100% of cells transfected with the positive control expressed both DsRed and acGFP. (**D**,**E**) In keratinocytes, correct *trans*-splicing to *COL7A1* was confirmed by RT-PCR using a GFP forward and a *COL7A1* exon 67 reverse primer. The 335 bp amplification product was verified to be the GFP-*COL7A1* fusion by Sanger sequencing.

**Figure 2 ijms-23-01732-f002:**
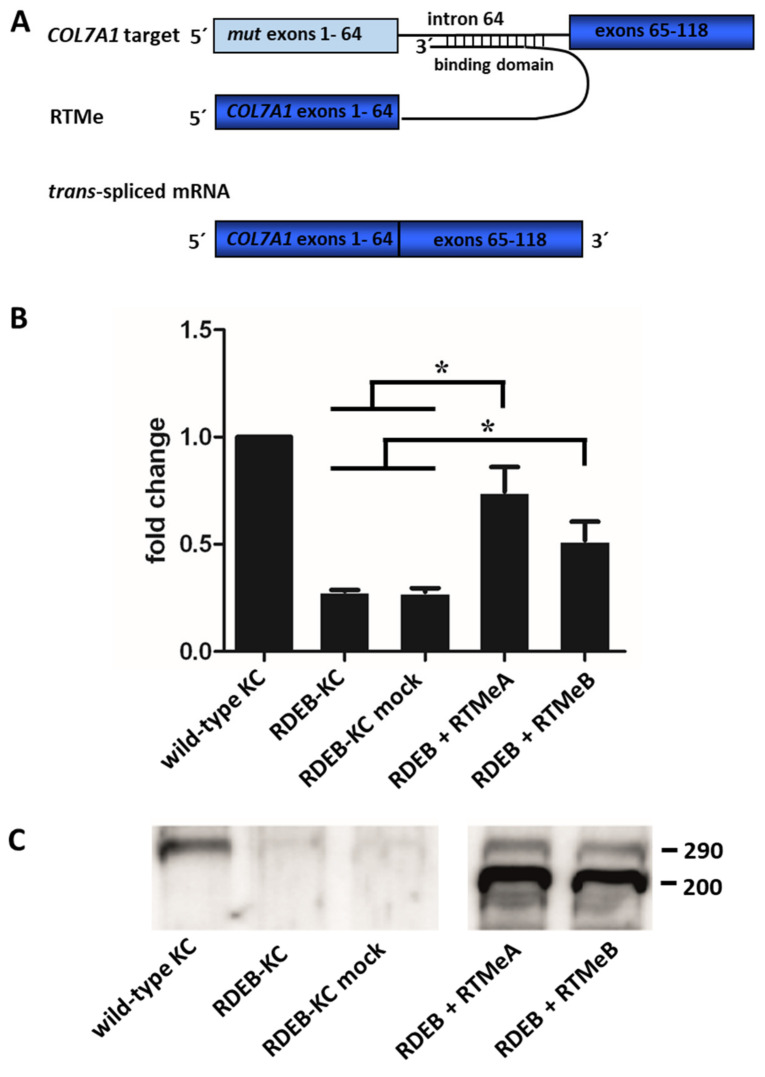
Endogenous *trans*-splicing: Gene and protein expression in RDEB-KC. (**A**) Schematic of the endogenous *trans*-splicing process. The complete transcript region from exon 1 to 64, which harbors the mutation of interest, is replaced by its wild-type copy provided by RTMe. (**B**) SqRT-PCR performed on an RDEB patient cell line retrovirally transduced with RTMeA and RTMeB. Means ± SD of fold changes of *COL7A1* expression over wild-type keratinocytes (KC) of three independent experiments are given. Unpaired, two-sided Student’s *t*-test was performed and *p*-values ≤ 0.05 were considered significant (*). (**C**) Western blot analysis showed re-expression and secretion of C7.

**Figure 3 ijms-23-01732-f003:**
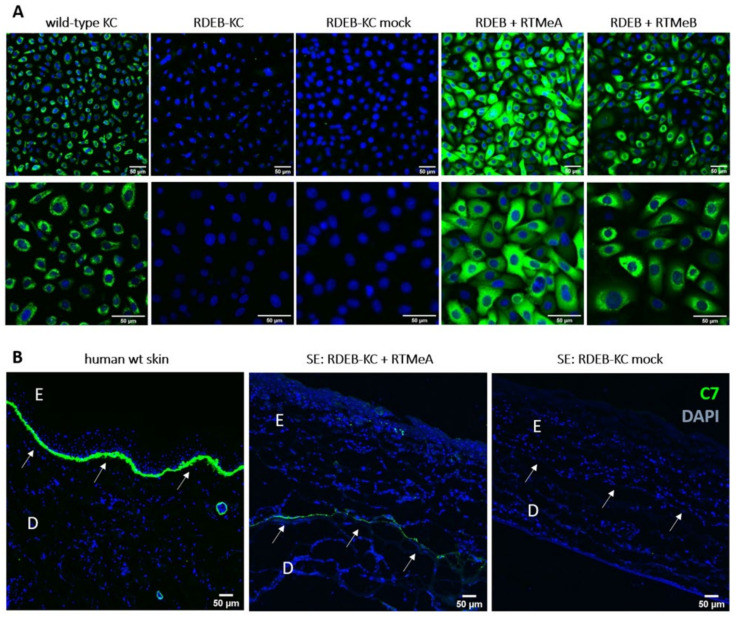
C7 immunostaining of RDEB-KC and skin equivalents. (**A**) C7-immunostaining (green) on RTM-transduced RDEB patient cells showed re-expression of C7. Cell nuclei were counterstained using DAPI (blue). (**B**) Correct localization of C7 at the BMZ was confirmed in cryosections of skin equivalents generated from RTMeA-transduced and mock-transduced (empty plasmid) patient KCs and in human wild-type (wt) skin. E = epidermis; D = dermis; arrows mark the dermal-epidermal junction. Scale bars: 50 μm.

## Data Availability

Data sharing not applicable as no datasets have been generated or analyzed in this study.

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
