# Peer review of "5′RNA Trans-Splicing Repair of COL7A1 Mutant Transcripts in Epidermolysis Bullosa"

_ijms, 2022, doi:10.3390/ijms23031732_

Round 1

Reviewer 1 Report

This paper by Mayr E et al. deals with the new approach to repair COL7A1 gene mutation with 5 end RNA trans splicing in RDEB keratinocytes. Previously, it has been demonstrated that 3 end RNA trans splicing can repair the mutation in COL7A1 gene. Here, the authors present that COL7A1 specific targeted RNA trans-splicing molecule at 5 end can also correct the mutation and restore COL7A1 expression in in vitro. This paper is well written and informative. However, I have the comments described below to improve this article.

Major comment is following:

1) About Figure1,

A: It is not clear that the mutation in COL7A1 gene in exon1-64 can be corrected in this system from this scheme. It is better to add the scheme of corrected COL7A1 product at the end.

B: Nuclear staining and merged (DsRed, GFP and nuclear staining) figures should be added. The efficacy of transduction also should be mentioned.

C: Is the ratio GFP positive cells divided by GsRed positive cells? The ratio is reversed in the paragraph (page3, line123 and 135). From the values shown in the graph (FACS), it is not able to recalculate the percentage mentioned (top on the FACS graphs). Please clarify these and explain well in the paragraph too.

D: There is no explanation on the sequence data on both RTMrs. Please indicate that exon number.

2)About Figure2

A: Please indicate which statistical test you used for and p-value in the figure legend.

B: It is better to add WB data from cell lysate too. Is there enough C7 in KC lysates too?

3)About Figure3

A: It is better to add the quantification of the staining (especially, comparison between WT-KC and transfected). This reviewer also strongly recommends to show the efficacy of transduction, otherwise it would be a cherry picking.

B: In the staining using skin equivalents, the direction of skin is not so clear. BMZ in RDEB-KC+RTM skin equivalents seems like way below than normal BMZ position. This suggests that proper and functional C7 might not be produced enough in this condition. Using other epidermis marker or BMZ marker staining would help to show BMZ more clearly. Moreover, HE staining to show the structure of skin, and electron microscopy figures to display  proper C7 are essential.

Minor comments are below:

1) In page10 line378 , the cell number is a typo. Please correct it.

Author Response

Point-by-Point Reply

Reviewer 1:

This paper by Mayr E et al. deals with the new approach to repair COL7A1 gene mutation with 5 end RNA trans splicing in RDEB keratinocytes. Previously, it has been demonstrated that 3 end RNA trans splicing can repair the mutation in COL7A1 gene. Here, the authors present that COL7A1 specific targeted RNA trans-splicing molecule at 5 end can also correct the mutation and restore COL7A1 expression in in vitro. This paper is well written and informative. However, I have the comments described below to improve this article.

Major comment is following:

1) About Figure1:

A: It is not clear that the mutation in COL7A1 gene in exon1-64 can be corrected in this system from this scheme. It is better to add the scheme of corrected COL7A1 product at the end.

We changed the title of Figure 1 to make clear that here the reporter setting is shown. In addition, we included a schematic showing the endogenous trans-splicing process in Figure 2A.

B: Nuclear staining and merged (DsRed, GFP and nuclear staining) figures should be added. The efficacy of transduction also should be mentioned.

We have performed the nuclear staining (Figure 1) and added the merged pictures in Supplementary Figure 1. The transfection efficiency is shown in the FACS blots in Figure 1C ranging from 8,3 to 90,6% considering that the target molecule carries no fluorescence reporter molecule.

C: Is the ratio GFP positive cells divided by GsRed positive cells? The ratio is reversed in the paragraph (page3, line123 and 135). From the values shown in the graph (FACS), it is not able to recalculate the percentage mentioned (top on the FACS graphs). Please clarify these and explain well in the paragraph too.

We have clarified the concern in the paragraph and in the Figure legend.

D: There is no explanation on the sequence data on both RTMrs. Please indicate that exon number.

We revised the Figure according to the concern. The binding domain sequences are described in paragraph 2.1. and 4.1.

2) About Figure 2:

A: Please indicate which statistical test you used for and p-value in the figure legend.

The respective information was included in the figure legend.

B: It is better to add WB data from cell lysate too. Is there enough C7 in KC lysates too?

We believe that the best indication for protein functionality is the protein level secreted in the supernatant. Here we see a clear difference between the non-treated and RTM-treated samples. Thus, we have not analysed the amount of C7 in the cell lysates.

3) About Figure3

A: It is better to add the quantification of the staining (especially, comparison between WT-KC and transfected). This reviewer also strongly recommends to show the efficacy of transduction, otherwise it would be a cherry picking.

Due to antibiotic selection we expect a transduction efficiency of ~100%. Although there is a possibility that truncated RTM variants are stable integrated into the genome, most RTM-treated keratinocytes showed an increased C7 staining. We have added new IF stainings in Figure 3.

B: In the staining using skin equivalents, the direction of skin is not so clear. BMZ in RDEB-KC+RTM skin equivalents seems like way below than normal BMZ position. This suggests that proper and functional C7 might not be produced enough in this condition. Using other epidermis marker or BMZ marker staining would help to show BMZ more clearly. Moreover, HE staining to show the structure of skin, and electron microscopy figures to display proper C7 are essential.

It is clearly visible that reduced amounts of C7 are located within the BMZ compared to the BMZ from skin equivalents expanded from wild-type human keratinocytes. However, we have performed a Laminin 332 staining (see Figure below), in which we see the same localization of Laminin 332 compared to C7 (Figure 3), and marked dermis and epidermis in Figure 3b. The performance of electron microscopy on skin equivalents is difficult, time-consuming, and therefore not manageable in this short revision time. However, we agree with the reviewer that this would be the best proof for functionality. Thus, we will consider that for future experiments.

Minor comments are below:

4) In page10 line378, the cell number is a typo. Please correct it.

The typo was corrected.

Reviewer 2 Report

Mayr et al present a study to demonstrate the capability of COL7a1 RTM to replace Col7a1 exons 1 to 64 in endogenous setting. The study is sound in design and controls but authors fail to emphasize the novelty of their work and what this study adds to the growing field of work on the topic. Following should be addressed before the study can be accepted for publication:

Introduction is too long, authors should more carefully emphasize the need for their study. 

Figure 1E labelled with 335bp on image and 334bp in Figure legend please correct.

Figure 3 include higher magnification images of cells.  

Figure 3A is it possible to quantify the data, i.e. analysis in 100 cells and in 3B quantification of fluorescence intensity/expression.

Is it possible to do EM on 3D skin equivalents to look at Col7a1 anchoring fibril structure or asses the quality/number of hemidesmosomes? At moment the paper lacks some more insight and evidence that would support the ICC/IHC. 

Discussion - please emphasize what is novel in abstract and conclusion. What are the next steps in this research?

Please include a section/statement on limitation of current study with suggestions of where the field needs to move to answer the remaining questions that are outstanding but outside the scope of this paper. 

Please check English throughout and avoid to start stntances with "So far..." and "It". 

Page 7, line 219 "decreased over time" - please be more specific about what the study found and how long did it take for Col7a1 to be decreased. 

Page 10 line 378 5 x 10^4 needs a correction. 4 should be superscript.

How much protein was loaded in WB and was the loading control run? 

Page 10 line 396 "About 3x 10^4 cells" please be specific remove "about". 

Section 4.12 Include a statement about ethics and patient consent for use of cells in research as primary cells were used.  

Author Response

Point-by-Point Reply

Reviewer 2:

Mayr et al present a study to demonstrate the capability of COL7a1 RTM to replace Col7a1 exons 1 to 64 in endogenous setting. The study is sound in design and controls but authors fail to emphasize the novelty of their work and what this study adds to the growing field of work on the topic. Following should be addressed before the study can be accepted for publication:

  • Introduction is too long, authors should more carefully emphasize the need for their study. 

We think it is interesting for the readers to get to know alternative gene therapeutic applications for dystrophic epidermolysis bullosa. Therefore, we mentioned the conventional full-length cDNA replacement therapy for DEB in more detail. The drawbacks of this therapeutic strategy justify the use of RNA trans-splicing as an alternative therapy option for the patient. We constitute the need of our study in the discussion section.

Figure 1E labelled with 335bp on image and 334bp in Figure legend please correct.

The figure legend was corrected.

  • Figure 3 include higher magnification images of cells.  

A: Figure 3A is it possible to quantify the data, i.e. analysis in 100 cells and in 3B quantification of fluorescence intensity/expression.

We repeated the IF stainings to show a higher number of treated cells in Figure 3. For quantification of C7 expression (e.g. via flow cytometric analysis) we would need more time for paper revision. Furthermore, we have included two different magnifications of the cells.

B: Is it possible to do EM on 3D skin equivalents to look at Col7a1 anchoring fibril structure or asses the quality/number of hemidesmosomes? At moment the paper lacks some more insight and evidence that would support the ICC/IHC. 

We agree with the reviewer that this would increase the impact of the study. However, the performance of electron microscopy on skin equivalents is difficult, time-consuming, and therefore not manageable in this short revision time. We will consider the suggestion of the reviewer for future in vitro and in vivo experiments.

  • Discussion - please emphasize what is novel in abstract and conclusion. What are the next steps in this research? Please include a section/statement on limitation of current study with suggestions of where the field needs to move to answer the remaining questions that are outstanding but outside the scope of this paper. 

Respective changes were made to the discussion section. Novelty is highlighted in abstract lines 25 - 27 and from lines 210 – 213 in the discussion section. Further strengths and limitations are now discussed from lines 253 - 258.

  • Please check English throughout and avoid to start stntances with "So far..." and "It". 

We have checked the English throughout the text.

  • Page 7, line 219 "decreased over time" - please be more specific about what the study found and how long did it take for Col7a1 to be decreased. 

We have added more details on the study and included information on the C7 half-life in the discussion section.

  • Page 10 line 378 5 x 10^4 needs a correction. 4 should be superscript.

This error war corrected.

  • How much protein was loaded in WB and was the loading control run? 

Our aim was to show that the C7 secretion was restored upon RNA repair. In general, we seeded the same amount of cells of the positive and negative control as well as RTM-treated cells. Then we collected the supernatants at certain time points. After blotting we have analysed the presence of rest protein via Coomassie staining (see Figure below).

  • Page 10 line 396 "About 3x 10^4 cells" please be specific remove "about". 

About was removed.

  • Section 4.12 Include a statement about ethics and patient consent for use of cells in research as primary cells were used.  

An ethics statement was included (lines 495 – 500).

Round 2

Reviewer 1 Report

The authors answered most of the raised questions in their first revision, but there is still unsolved issue remained.

Major comment is following:

3)About Figure3

B: The authors described that BMZ position is correct in treated RDEB KCs derived skin equivalents, but they did not provide other data about it. In point to point response, they mentioned Laminin332 staining colocalized with C7, so this data should be added. At least HE staining is required to show that proper epidermal structure is formed in that skin equivalent. In addition, DAPI intensity in skin equivalents of both conditions are lower than human wt skin, so the structure of them are not well observed. Please make them uniform.

Author Response

Point-by-Point Reply

Reviewer I:

The authors answered most of the raised questions in their first revision, but there is still unsolved issue remained.

Major comment is following:

3) About Figure3

B: The authors described that BMZ position is correct in treated RDEB KCs derived skin equivalents, but they did not provide other data about it. In point to point response, they mentioned Laminin332 staining colocalized with C7, so this data should be added. At least HE staining is required to show that proper epidermal structure is formed in that skin equivalent. In addition, DAPI intensity in skin equivalents of both conditions are lower than human wt skin, so the structure of them are not well observed. Please make them uniform.

We thank the reviewer for the valuable comments. We have addded now the co-staining of type VII collagen and laminin 332 as Supplementary Figure 2 showing a clear colocalization of both proteins within the BMZ of generated skin equivalents derived from corrected RDEB keratinocytes. Furthermore, we adapted the DAPI staining in Figure 3.

Reviewer 2 Report

Authors have addresses most minor issues of the suggestions however additional experimental data to support their original claims is still lacking. Therefore I do not believe the submission has changed significantly. I do not have any further comments to provide. 

Author Response

We thank the reviewer for the valuable comments.

Round 3

Reviewer 1 Report

The authors did not show H&E of skin equivalent. However it is clear, from  multiple imunnostaining, that while the deposition of C7 is restored, basal cells can not bind properly to the basal membrane in treated skin equivalent. This suggest that the treatment is not completly functional. So, at least this should be acknowledged in the manuscript.

Reviewer 2 Report

The authors have addressed most of my concerns and I believe this paper should now be accepted for publications as it has significantly improved from the original submission. Well done.